# Comparison of Microbial Sampling Sites and Donor-Related Factors on Corneal Graft Contamination

**DOI:** 10.3390/jcm11216236

**Published:** 2022-10-22

**Authors:** Yu-Jen Wang, Ko-Chiang Sung, Wei-Chen Lin, Fu-Chin Huang

**Affiliations:** 1Department of Parasitology, School of Medicine, China Medical University, Taichung 404333, Taiwan; 2Department of Clinical Laboratory, Chest Hospital, Ministry of Health and Welfare, Tainan 71742, Taiwan; 3Department of Parasitology, College of Medicine, National Cheng Kung University, Tainan 70101, Taiwan; 4Department of Microbiology and Immunology, College of Medicine, National Cheng Kung University, Tainan 70101, Taiwan; 5Department of Ophthalmology, National Cheng Kung University Hospital, College of Medicine, National Cheng Kung University, Tainan 70403, Taiwan

**Keywords:** cornea donation, contamination, microbial sampling

## Abstract

Grafts used for corneal donation should be sterile to avoid transplantation failure and secondary infection. However, there are no clear and globally accepted specifications from eye banks on microbial sampling sites. The objective of this study was to analyze microbial contamination of corneal grafts collected from different sampling sites. We found that the contamination rates and strain compositions significantly differed at different sampling sites. To clarify the effect of the microbial sampling site on corneal graft contamination, microbial sampling was conducted using 30 corneal grafts at the extraocular and intraocular sides of the graft in 2020 from the National Eye Bank of Taiwan. Microbial contamination significantly differed (*p* < 0.05) between the different sampling sites on the graft according to McNemar’s test. Although the two sampling sites showed the same specificity (33.33%), the sensitivity of sampling on the extraocular side (82.35%) was higher than that on the intraocular side (17.65%) of the graft. Donor-associated factors, including the cause of death, operating place, and cold compression, were analyzed using chi-square statistics, which revealed no significant differences in microbial contamination. Thus, our data provide evidence for the microbial sampling site of donated grafts and clear specifications for maintaining the quality of corneal grafts.

## 1. Introduction

Corneal transplantation can be performed in patients suffering from irreparable corneal damage to restore vision. However, complications can occur after corneal transplantation, including endophthalmitis as the most severe condition. Transplantation of a contaminated cornea was the most common cause of post-transplant endophthalmitis and can cause blindness and loss of the eye [1,2]. Several methods have been used to reduce the risk of contamination of donor corneas, such as ophthalmological assessment of corneal donors, decontamination procedures before corneal excision, graft preservation under sterile conditions, and antibiotic prophylaxis in the medium [3]. However, contamination rates of corneal grafts remain high at approximately 9.97% [4,5,6]. Several factors, such as donor’s age and sex, causes of death, and different procurement and preservation techniques, may impact donor corneal contamination [7,8]. According to a previous study, the most common risk factors for contamination include a donor who died of sepsis and the time between death and donation [9]. However, the correlation between contamination rate and microbial sampling between donation surgical procedures has not been reported.

To date, the standard operating procedures of eye banks in various countries do not provide clear guidance on the microbial sampling sites on grafts. The Eye Bank Association of America and the Advisory Committee on the Safety of Blood, Tissues, and Organs in the United Kingdom only proposed that an environment in which corneal tissue is processed should contain acceptable levels of airborne microbial contamination [10]. Although microbial contamination of donor eyes does not always result in infection, and pre-surgical or surgical cultures may not correlate with postoperative infection, if it occurs, the Eye Bank Association of India suggests that grafts can be cultured before surgery and/or during surgery [11]. According to the standard operating procedure of the National Eye Bank of Taiwan (NEBT), donors with contraindications are excluded from donation. These donors include those with human immunodeficiency virus, syphilis, hepatitis C virus, human T-lymphotropic virus 1, active fungemia, and active ocular inflammation. Microbial sampling during surgery can be performed using a culture swab to inspect the corneoscleral rim, but the sampling site guidelines are unclear.

In this study, the graft contamination rate and microbial strain composition of previous cases, collected from different sampling sites, were analyzed. Microbes identified from the sampling site on the extraocular and intraocular sides of the corneoscleral rim of samples from 30 participants were compared, and donor-associated factors were analyzed based on the microbial sampling results. 

## 2. Materials and Methods

### 2.1. History and Contamination Rate Analysis

From 2016 to 2019, consecutive contaminated corneal grafts from the southern office of the NEBT were analyzed to determine the microbial strain composition. All corneal grafts were harvested by the same staff team and using the same standard operating procedure in the southern office of the NEBT. Microbiological tests were performed by the Department of Pathology at the National Cheng Kung University Hospital.

### 2.2. Donor-Associated Factor Collection

Factors associated with corneal donors were extracted from the NEBT in the southern office. This study was conducted in accordance with the guidelines of the Declaration of Helsinki and was approved by the Institutional Review Board of the National Cheng Kung University Hospital (IRB No: B-ER-109-108).

### 2.3. Donation Surgery and Microbial Sampling

The donation surgery procedures were performed according to the manual of standard operating procedures in NEBT, and 30 corneal grafts were harvested by the same team in the southern office of NEBT. Briefly, the dominant ocular commensal was inspected with a sterile cotton swab on the conjunctiva of the donor before excising the corneoscleral rim. The donor’s eye was rinsed with physiological NaCl solution, followed by conjunctival and cutaneous antisepsis procedures using 5% povidone-iodine with a contact period of 2 min. The corneoscleral rim was removed under sterile surgical conditions, and the corneal limbus of the extraocular and intraocular sides of the sclera was sampled using a sterile cotton swab. All sampling swabs were immediately preserved in lysogeny broth in a sterile culture tube, and the corneal grafts were stored at 4 °C in Optisol GS media (Bausch & Lomb Surgical, Irvine, CA, USA) in a viewing chamber (Alcon Surgical, Chicago, IL, USA).

### 2.4. Microbiological Identification

To identify the donor’s dominant ocular commensal, the sampling swab was preserved in lysogeny broth and transported to the laboratory at 4 °C. The sample containing the swab was cultured in a shaking incubator at 180 rpm and 35 °C (Yihder, New Taipei, Taiwan). After overnight incubation, bacterial pellets were suspended in distilled water for genome extraction. DNA was extracted using a LabPrep DNA Mini Kit ^®^ (TAIGEN Bioscience Corporation, Taipei, Taiwan) according to the manufacturer’s instructions. The enriched DNA samples were subjected to polymerase chain reaction to amplify the V3-V4 genomics region of the bacterial 16S rRNA gene using specific forward (TCGTCGGCAGCGTCAGATG TGTATAAGAGACAGCCT ACGGGNGGCWGCAG) and reverse primers (GTCTCGTGGGCTCGGAGATGTG TATAAGAGACAGGACTACHVGGGTATCTAATCC), as described in our previous study [12]. The amplified DNA samples were sequenced at the National Center for Biotechnology Information (NIH, Bethesda, MD, USA) to identify the microbial strains.

### 2.5. Statistical Analysis

Swab culture data were compared using descriptive analysis. The sampling method was compared using McNemar’s test to calculate the significance. Sensitivity was calculated using the following formula: number of true-positives/number of true-positives + number of false-negatives. Specificity was calculated using the following formula: number of true-negatives/number of true-negatives + number of false-positives. Donor-associated factors were evaluated using the chi-square test based on the microbiological test results. Statistical analyses were conducted using the SAS statistical software (version 9.4; SAS Institute, Cary, NC, USA). The statistical data were illustrated using the GraphPad Prism version 5.0 software (La Jolla, CA, USA).

## 3. Results

### 3.1. Changes in Contamination Rate and Microbial Strain Composition after Modification of the Microbial Sampling Site

To determine whether the corneal graft contamination rate was affected by the microbial sampling site, the corneal graft contamination rate and microbial strain composition were evaluated from 2016 to 2019. The microbial sampling site was changed from the extraocular side of the corneoscleral rim in 2016 and 2017 to the intraocular side of the corneoscleral rim in 2018 and 2019. Unexpectedly, the graft contamination rate greatly decreased from 16.15% and 14.87% to 7.96% and 8.63%, respectively, after changing the microbial sampling site (Figure 1). The operating staff and procedures were not changed during this period, indicating that the decrease in the graft contamination rate was caused by the change of the microbial sampling site. We further analyzed and screened the microbial strain composition from the positive cases detected during the microbiological tests. The microbial strain composition revealed 11 and 12 microbes in 2016 and 2017, respectively. In contrast, five and six microbes were detected in 2018 and 2019, respectively (Figure 2). The contamination rate and microbial stain composition in 2016–2017 and 2018–2019 suggests that the different microbial sampling sites at the corneoscleral rim influenced graft contamination.

### 3.2. Comparison of Microbial Sampling Site and Classification of Donor-Associated Factors

According to the statistical results of previous graft contamination, we evaluated 30 corneal grafts to compare the microbial sampling sites and donor-associated factors. Corneal donors were collected from subjects from central and southern Taiwan who had no infectious diseases, according to the NEBT policies. We examined the ocular surface commensals before surgery, contaminating microbes at the extraocular side of the corneoscleral rim, and contaminating microbes at the extraocular side of the corneoscleral rim (Figure 3). Pre-surgery microbial sampling was performed to determine the true- or false-negative microbial test results in post-surgery sampling. The post-surgery sampling results were compared for consistency. The contamination rate of the extraocular and intraocular sides of the corneoscleral rim was further analyzed to determine the influence of donor-associated factors. The compared factors included: death caused by an accident (subarachnoid hemorrhage, subdural hematoma, intracerebral hemorrhage, and intraventricular hemorrhage); internal disease (chronic pulmonary embolism, acute myocardial infarction, and cerebrovascular accident); cancer (rectal cancer, lung cancer, tongue cancer, pancreatic tail adenocarcinoma, sigmoid adenocarcinoma, and esophageal cancer); operation in the operating room, intensive care unit, or contamination zone (ward or emergency room); and immediate cold compression (Figure 4). Donor-associated factors were compared in order to exclude factors interfering with the graft contamination results.

### 3.3. Sampling at the Extraocular Side of the Corneoscleral Rim Revealed Higher Sensitivity than That at the Intraocular Side

McNemar’s test was performed to compare the consistency in the microbiological test results between the extraocular and intraocular microbial sampling sites of the corneal grafts. Thirty corneal grafts were sampled post-operatively from the extraocular and intraocular sides of the corneoscleral rim. On the extraocular side of the graft, 16 grafts (53.33%) were contaminated and 14 grafts (46.67%) were sterile. On the intraocular side of the graft, 5 grafts (16.67%) were contaminated and 25 grafts (83.33%) were sterile. These results revealed significant differences (*p* < 0.05) in the microbiological test results between the two different microbial sampling sites of the corneal graft (Table 1). The sensitivity and specificity of the two microbial sampling procedures were calculated. When using the same method for identifying the microbial strains, the specificities were equal for the two sampling sites (33.33%). However, microbial sampling at the extraocular side of the corneal graft showed a higher sensitivity (82.35%) than that of intraocular side sampling (17.65%) (Table 2).

To reveal the microbial sampling availability, a graft showing negative results on both sampling sides was considered as completely disinfected. Among the 20 contaminated corneal grafts, only 4 cases (20%) were not detected by sampling at the extraocular side of the corneoscleral rim; however, 15 cases (75%) were not detected by sampling at the intraocular side of the corneoscleral rim. Although a small number of cases were evaluated, more microbial strains were detected on the extraocular side than on the intraocular side of the graft (Figure 5). These results indicate that microbial sampling at the extraocular side of the graft had a higher sensitivity and better identification ability than that at the intraocular side of the graft for analyzing graft contamination.

### 3.4. Difference in Sampling Results Is Not Caused by Donor-Associated Factors

Finally, we performed the chi-square test to analyze donor-associated factors including cause of death, operating location, and cold compression time according to the microbiological test results at the two sampling sites (Table 3). The cause of death (accident, internal disease, and cancer) was not significantly associated with the microbiological test results on the extraocular (*p* = 0.251) or intraocular (*p* = 0.392) sampling sides. The same results were observed for the operating location and timing of cold compression. The contamination zone was defined as ward and emergency room and the clean zone as the intensive care unit and operating room; these locations were not associated (*p* = 0.892; *p* = 0.387) with the microbiological test results. Similarly, cold compression was performed to slow microbial growth with no interference (*p* = 0.215; *p* = 1.000) at the sampling sites, whether immediately or not. The data suggest that the donor’s cause of death, donation location, and cold compression on the eyes did not interfere with the microbiological test results at either microbial sampling site.

## 4. Discussion

A previous study suggested that routine microbiological screening of corneal grafts is necessary to prevent endophthalmitis after penetrating keratoplasty [13]. However, the results of microbial sampling testing of corneal grafts differ between eye banks worldwide. An eye bank in France performs testing on day 5 in a transport medium [14]. Hermel et al., used an automated test detection system to investigate the optimal time for sterility testing [15]. Sterile swabs were used to inspect the donor’s upper and lower conjunctiva for microbial detection after povidone-iodine disinfection [16]. Thus, microbial sampling for corneal grafts is not standardized and remains a controversial issue. In Taiwan, we collected corneal graft samples post-surgery for microbiological testing. Graft preservation and microbiological identification tests were performed concurrently after surgery. The preventive antibiotics penicillin and streptomycin were added to the medium, and the microbiological test results were reported within 3 days. The microbial colony in the positive microbiological test was further identified in an antibiotic sensitivity test. However, whether dead microbial cells in preserved medium affect the graft quality remains unclear.

*Staphylococcus* spp. are the most common bacteria isolated from corneal grafts after disinfection [17] and the dominant ocular commensal of human eyes. Furthermore, recent studies have revealed that *Staphylococcus aureus* and coagulase-negative *Staphylococcus* make up the majority of the human ocular microbiota [18,19], and *Staphylococcus* spp. has also been the focus of research on skin colonization [20]. Notably, the ocular surface and skin microbiota are common sources of graft contamination that may be caused by incomplete disinfection and surgical contamination, and those bacteria are also the predominant isolates in our study. However, in most eye banks, the major source of corneal grafts is hospitalized patients. The drug-resistant *Staphylococcus* spp. has recently become the most common pathogen in nosocomial infections [21], suggesting that hospitalized donors have a higher frequency of graft contamination by drug-resistant *Staphylococcus* spp. Hence, accurately detecting these drug-resistant bacteria is important for preventing post-transplant endophthalmitis. Although some contaminating pathogens are killed by antibiotics in preserved medium, the actual number of contaminated colonies is unknown. Several Gram-negative bacteria such as *Pseudomonas aeruginosa* and *E. coli* have been isolated from contaminated corneal grafts. Endotoxins are major constituents of the outer cell [22]. When Gram-negative bacteria are treated with antibiotics, endotoxin release occurs at various levels [23,24]. Therefore, effective microbial sampling of corneal grafts can detect drug-resistant bacterial contamination and predict the number of contaminating bacteria for graft quality maintenance.

Microbial sampling is a widely used method in surgical studies. One dental study showed that bone debris should be collected for bacterial identification [25]. Operating room air is considered as a risk factor for contamination during total joint arthroplasty [26]. Cleaning of surgical instruments has been suggested to prevent contamination during surgery [27]. However, the anatomical position that should be used for surgical microbial sampling is not typically mentioned in manuals and guidelines. Although microbiological cultivation was used in our study to demonstrate the differences in the sensitivity and specificity of the sampling sites, since the number of ocular surface and surgically contaminated bacteria are often lower than that in other parts of the human body, future related studies could further use molecular assays to explore bacteria that may be missed in culture. Moreover, donated tissues and organs are limited in Asia, particularly in Taiwan. Serious contamination causes not only post-transplant infection but also rare graft wastage. Hence, we investigated the differences in the location of microbial sampling and showed that institutions should strictly strengthen their process specifications to improve the quality of donated corneas.

## 5. Conclusions

In this study, we compared two different microbial sampling sites on the corneoscleral rim; we found that the extraocular side of corneal graft sampling exhibited a higher sensitivity than the intraocular side of corneal graft sampling for contamination identification. To the best of our knowledge, this is the first study to compare microbial sampling of the corneal grafts. Microbial sampling should be performed on the extraocular side of the corneoscleral rim to detect contamination from operators or failed disinfection. These factors should be clearly described in manuals of standard operating procedures. Finally, our results provide information that can guide microbial sampling of the corneal graft and improve the quality of cornea donation.

## Figures and Tables

**Figure 1 jcm-11-06236-f001:**
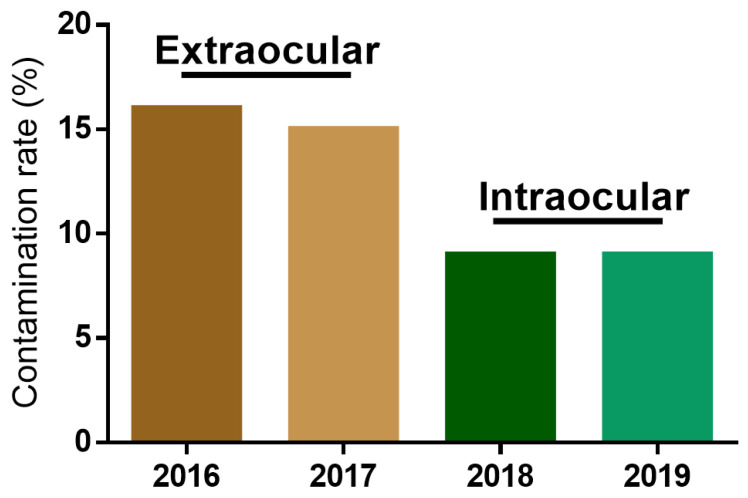
Corneal graft contamination rates in 2016–2019. Microbial sampling was performed at the extraocular side of the corneoscleral rim from 2016 to 2017 and at the intraocular side of the corneoscleral rim from 2018 to 2019. The corneal graft contamination rates were: 16.15% in 2016, 14.87% in 2017, 7.96% in 2018, and 8.63% in 2019.

**Figure 2 jcm-11-06236-f002:**
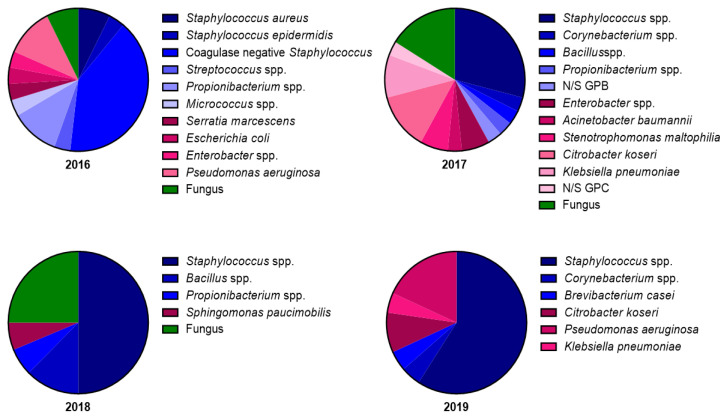
Composition statistics of microbial strains in cases positive for microbial contamination. We identified 11 and 12 microbial strains in 2016 and 2017, respectively, from the extraocular side of corneoscleral rim sampling. We identified 5 and 6 microbial strains in 2018 and 2019, respectively, from the intraocular side of corneoscleral rim sampling. N/S, non-specific; GPB, Gram-positive bacillus; GNC, Gram-negative cocci.

**Figure 3 jcm-11-06236-f003:**
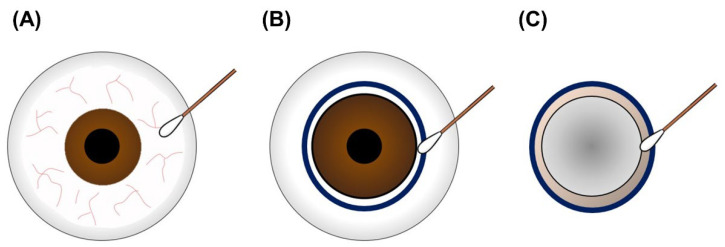
Microbial sampling sites in this study. All microbial sampling was performed using a sterile cotton swab, and the samples were preserved in autoclaved lysogeny broth. (**A**) Pre-surgery microbial samples were collected from the conjunctiva of the ocular surface. Post-surgery samples were collected from the extraocular corneal limbus (**B**) and intraocular corneoscleral rim (**C**) for surgical contamination analysis.

**Figure 4 jcm-11-06236-f004:**
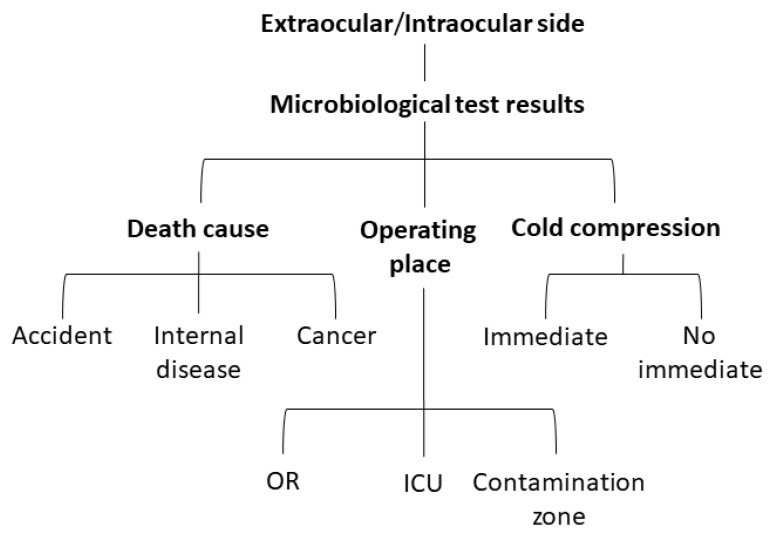
Group comparison of donor-associated factors with microbiological test results. The microbiological test results of the extraocular or intraocular side at the corneoscleral rim were also compared with three donor-associated factors including the cause of death, operating location, and cold compression time. OR, operating room; ICU, intensive care unit.

**Figure 5 jcm-11-06236-f005:**
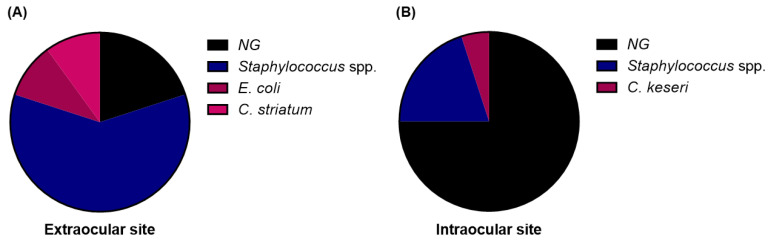
Comparison of undetected rate and abundance of detected microbial strains by sampling site at the extraocular and intraocular sides of the contaminated graft. (**A**) Extraocular side sampling revealed 12 cases (60%) of *Staphylococcus* spp., 2 cases (10%) of Escherichia coli, 2 cases (10%) of *Corynebacterium striatum*, and 4 undetected cases (20%). (**B**) Extraocular side sampling revealed 4 cases (20%) of *Staphylococcus* spp., 1 case (5%) of *Citrobacter koseri*, and 15 undetected cases (75%). NG, no growth.

**Table 1 jcm-11-06236-t001:** Comparison of microbiological test results between microbial sampling of the extraocular and intraocular corneoscleral rims.

Intraocular Site				
		−	+	*p*-Value
Extraocular site	−	10 (33.33)	4 (13.33)	0.019 *
	+	15 (50.00)	1 (3.33)	

Data presented as frequency (%); * *p*-value from chi-squared tests; No significant: *p* > 0.05.

**Table 2 jcm-11-06236-t002:** Method sensitivity and specificity of the microbial sampling site at the extraocular and intraocular sides of the corneoscleral rim.

**Extraocular Site**			
		Pre-surgery sampling	
		(+)	(−)
Post-surgery sampling	(+)	14 (70)	2 (10)
	(−)	3 (15)	1 (5)
Sensitivity: 82.35% Specificity: 33.33%			
**Intraocular site**			
		Pre-surgery sampling	
		(+)	(−)
Post-surgery sampling	(+)	3 (15)	2 (10)
	(−)	14 (70)	1 (5)
Sensitivity: 17.65% Specificity: 33.33%			

**Table 3 jcm-11-06236-t003:** Correlation between the Donor-Associated Factors and Microbiological Test Results at the Extraocular and Intraocular Sides of the Graft.

	**Extraocular Site**		
	−	+	*p*-Value
	(*n* = 14)	(*n* = 16)	
	*n* (%)	*n* (%)	
**Death cause**			
Accident	6 (42.86)	4 (25.00)	0.251
Cancer	6 (42.86)	5 (31.25)	
Internal disease	2 (14.29)	7 (43.75)	
**Operating place**			
Contamination zone	3 (21.43)	4 (25.0)	0.892
ICU	3 (21.43)	5 (31.25)	
OR	8 (57.14)	7 (43.75)	
**Cold compression**			
No immediate	3 (21.43)	8 (50.00)	0.215
Immediate	11 (78.57)	8 (50.00)	
	**Intraocular Site**		
	−	+	*p*-Value
	(*n* = 25)	(*n* = 5)	
	*n* (%)	*n* (%)	
**Death cause**			
Accident	9 (36.00)	1 (20.00)	0.392
Cancer	10 (40.00)	1 (20.00)	
Internal disease	6 (24.00)	3 (60.00)	
**Operating place**			
Contamination zone	5 (20.00)	2 (40.00)	0.387
ICU	8 (32.00)	0 (0.00)	
OR	12 (48.00)	3 (60.00)	
**Cold compression**			
No immediate	9 (36.00)	2 (40.00)	1.000
Immediate	16 (64.00)	3 (60.00)	

Data presented as frequency (%); *p*-value from chi-squared tests; No significant: *p* > 0.05.

## Data Availability

Data sharing is not applicable to this article.

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
