# Peer review of "Comparison of Microbial Sampling Sites and Donor-Related Factors on Corneal Graft Contamination"

_jcm, 2022, doi:10.3390/jcm11216236_

Round 1
Reviewer 1 Report
The authors have conducted a well organized study on the results of corneal eye banking microbiology.
The following points need to be addressed:
- It should be clarify that what exactly means " Lateral" versus " Medial"; Initially one may think of Nasal conjunctiva versus Temporal conjunctiva.
In my opinion better to say " extraocular side" vs " intraocular side".
- It is not surprise that more positive culture results to be expected from The samples taken from The extraocular surface rather than intraocular side. Author need to add more correlations on this to make it more attractive to the readers. For example finding a correlation between the culture results and normal microbial flora of the eyelid Margins.
Reviewer 2 Report
Authors wrote an interesting article
Few improvements are needed
Introduction:
Please have a look and cite the following paper: PMID: 31179394
Please write a couple o sentences regarding metagenomics and its potentials (PMID: 36142138)
Methods: please expand the method and make them more clear to understand
Please expand the limitations of the study, especially the limitations given by CDM and culture
Otherwise it is a good article
Round 2
Reviewer 1 Report
Thanks for the reply.
Looks more reasobnable with the corrections.
Author Response
Thank you very much for your helpful suggestions and corrections. It is hoped that this study can provide a basis for the standardization of microbial test in transplant operations, so as to improve the quality of tansplant grafts.